# Three Small Cysteine-Free Proteins (CFP1–3) Are Required for Insect-Pathogenic Lifestyle of *Metarhizium robertsii*

**DOI:** 10.3390/jof8060606

**Published:** 2022-06-06

**Authors:** Ya-Ni Mou, Kang Ren, Si-Yuan Xu, Sheng-Hua Ying, Ming-Guang Feng

**Affiliations:** Institute of Microbiology, College of Life Sciences, Zhejiang University, Hangzhou 310058, China; 11907036@zju.edu.cn (Y.-N.M.); 11807120@zju.edu.cn (K.R.); 12007036@zju.edu.cn (S.-Y.X.); yingsh@zju.edu.cn (S.-H.Y.)

**Keywords:** Hypocreales, asexual insect pathogens, cysteine-free proteins, gene expression and regulation, insect pathogenicity, insecticidal activity

## Abstract

Unique CFP (cysteine-free protein; 120 aa) has been identified as an extraordinary virulence factor in *Beauveria bassiana* (Cordycipitaceae), a main source of wide-spectrum fungal insecticides. Its homologs exclusively exist in wide-spectrum insect pathogens of Hypocreales, suggesting their importance for a fungal insect-pathogenic lifestyle. In this study, all three CFP homologs (CFP1–3, 128–145 aa) were proven essential virulence factors in *Metarhizium robertsii* (Clavicipitaceae). Despite limited effects on asexual cycles in vitro, knockout mutants of *cfp1,*
*cfp2* and *cfp3* were severely compromised in their capability for normal cuticle infection, in which most tested *Galleria mellonella* larvae survived. The blocked cuticle infection concurred with reduced secretion of extracellular enzymes, including Pr1 proteases required cuticle penetration. Cuticle-bypassing infection by intrahemocoel injection of ~250 conidia per larva resulted in a greater reduction in virulence in the mutant of *cfp1* (82%) than of *cfp2* (21%) or *cfp3* (25%) versus the parental wild-type. Transcriptomic analysis revealed dysregulation of 604 genes (up/down ratio: 251:353) in the Δ*cfp1* mutant. Many of them were involved in virulence-related cellular processes and events aside from 154 functionally unknown genes (up/down ratio: 56:98). These results reinforce the essential roles of small CFP homologs in hypocrealean fungal adaptation to insect-pathogenic lifestyle and their exploitability for the genetic improvement of fungal insecticidal activity.

## 1. Introduction

*Beauveria* (Codycepitaceae) and *Metarhizium* (Clavicipitaceae) in Hypocreales are insect-pathogenic fungi deficient of teleomorph and serve as the main sources of environment-friendly mycoinsecticides [1,2]. Fungal insecticidal activity is an overall output of various cellular processes and events involved in fungus-insect interactions. Insect-pathogenic fungal genomes comprise approximately 15% genes involved in fungal infection and kill action and hence are speculated to be virulence factors [3,4,5,6,7]. Among hundreds of genes characterized in the post-genomic era, however, virulence factors exploitable for genetic improvement of fungal insecticides remain limited [8,9,10].

Fungal insect pathogenicity and virulence are distinct, but related, concepts in a general sense [11]. After conidial germination on insect surfaces, hyphal invasion into the insect body begins under the actions of extracellular (secreted) enzymes involved in proteolysis, chitinolysis and lipolysis [12,13,14], such as subtilisin-like Pr1 family proteases collectively required for cuticular penetration [15], and/or of the mechanic pressure of appressoria formed at the hyphal tips. As specialized infective structures, appressoria often form in the infection course of *Metarhizium* spp. [16,17] instead of *Beauveria bassiana* [18], implicating somewhat distinctive mechanisms of the two fungal lineages in the host infection. Upon arrival in the host hemocoel, hyphae must overcome the host immune defense responses [4,13] before the formation of hyphal bodies (thin-wall blastospores), which proliferate in vivo by yeast-like budding to accelerate mycosis development and host mummification to death [19,20,21]. Thus, fungal insect pathogenicity and virulence rely upon the capabilities of normal infection through cuticular penetration and subsequent hemocoel colonization, respectively.

Many small proteins, particularly cysteine-rich proteins, secreted in the host infection course of plant-pathogenic fungi, are identified virulence factors [22,23,24], such as multiple Avr proteins targeted by resistant proteins in the host plant [25,26] and extracellular proteins to stimulate a hypersensitive response of the host plant [27,28]. Such small secreted proteins (SSPs) are also speculated as a large class of key virulence factors in fungal insect pathogens [3,13] despite a scarcity of direct evidence. Revealed by previous time-course transcriptome analyses, hundreds of secreted and non-secreted protein genes were highly induced in the first 48-h infection course of cuticular penetration and hemocoel colonization by *B. bassiana* against *Plutella xylostella* [29]. Some of those genes have been identified as virulence factors due to sharply reduced or even abolished pathogenicity in the absence of each gene [30,31,32,33,34]. Among the studied genes, only one encodes small cysteine-rich protein [13 Cys resides in 230 amino acids (aa)] but proved a vacuole-localized protein (VLP4) [30]. In a recent study to characterize 10 SSP genes in *B. bassiana*, only one encoding small cysteine-free protein (CFP, 120 aa) was identified as an extraordinary virulence factor, and the remaining SSP genes contributed little to virulence irrespective of their encoding cysteine rich or deficient proteins [35]. Overexpression of this CFP gene resulted in a marked increase in the fungal insecticidal activity [36]. The previous studies indicate that cysteine richness is not necessarily a criterion for SSPs to be considered as virulence factors in fungal insect pathogens. Revealed by genome survey, one to three CFP homologs (96–145 aa) exist exclusively in six wide-spectrum insect pathogens of Codycepitaceae (*Beauveria brongniartii* and *Cordyceps fumosorosea*) and Clavicipitaceae (*Metarhizium anisopliae*, *M.*
*brunneum*, *M. guizhouense* and *M. robertsii*) [35]. However, no homolog was found in other ascomycetes, including entomopathogenic (*Ascosphaera api*, *C.*
*militaris*, *C. confragosa*, *Moelleriella libera* and *Sporothrix insectorum*) or non-entomopathogenic (*Aspergillus*, *Botrytis*, *Fusarium*, *Neurospora* and *Pyricularia*) fungi. Neither was any CFP homolog found in the specialists *M. acridum* (Orthoptera) and *M.*
*rileyi* (Lepidoptera). Since the six insect pathogens are all important sources of fungal pesticides [1], their CFP homologs are possible virulence factors to be exploited for improving the insecticidal activities of those fungi as does CFP in *B. bassiana*. This study sought to test the hypothesis by functional analysis of three CFP homologs (named CFP1–3) in *M. robertsii*. As presented below, CFP1–3 were evidently required for insect pathogenicity and virulence of *M. robertsii*, and the most important CFP1 was involved in transcriptional mediation of hundreds of genes associated with virulence-related cellular events.

## 2. Materials and Methods

### 2.1. Recognition and Sequence Analysis of CFP Homologs in M. robertsii

The amino acid sequence of unique CFP (EJP69088) studied in *B. bassiana* [35,36] was used as a query to search through the *M. robertsii* genome [4] at http://blast.ncbi.nlm.nih.gov/blast.cgi/ (accessed on 4 June 2022). The identified homologs were compared with the query through sequence alignment analysis at http://www.bio-soft.net/format/DNAMAN.htm/ (accessed on 4 June 2022) and conserved domain analysis at http://smart.embl-heidelberg.de/ (accessed on 4 June 2022). Signal peptide and nuclear localization signal (NLS) motif were predicted from each homolog sequence at http://www.cbs.dtu.dk/services/SignalP-4.1/ (accessed on 4 June 2022) and https://www.novopro.cn/tools/nls-signal-prediction.html/ (accessed on 4 June 2022), respectively.

### 2.2. Generation of Targeted Gene Mutants

The nucleotide sequences of *cfp1* (447 bp), *cfp2* (489 bp) and *cfp3* (509 bp) were disrupted by deleting the promoter/coding regions of 438 (46 + 392), 459 (75 + 384) and 362 (16 + 346) bp from the genome of the wild-type strain *M. roberstii* ARSEF 2575 (designated WT), respectively (Appendix A). The promoter/coding region of each target gene was deleted by homologous recombination of 5′ and 3′ flanking/coding fragments separated by *bar* marker in the constructed vector p0380-5′*x*-bar-3′*x* (*x* = *cfp1*, *cfp2* or *cfp3*); the deleted region was complemented into an identified deletion mutant (DM) of each gene by ectopic integration of a cassette composed of full-length coding and flanking sequences and *sur* marker in the vector p0380-*x*-sur. The two vectors of each target gene were constructed and integrated into the respective WT and DM strains via *Agrobacterium* mediated transformation, as described previously [35]. Putative mutant colonies screened using *bar* resistance to phosphinothricin (200 μg/mL) or *sur* resistance to chlorimuron ethyl (10 μg/mL) were identified via PCR and real-time quantitative PCR (qPCR) analyses. All PCR and qPCR primers used for DNA amplification, vector construction and targeted gene detection are listed in Appendix A. The identified DM and CM strains (Appendix A) were evaluated together with the WT strain in the experiments, which included three independent replicates to meet a requirement for one-factor (strain) analysis of variance and Tukey’s honestly significant difference (HSD) test for phenotypic differences between DM and control (WT and CM) strains.

### 2.3. Assessments of Growth Rates under Different Culture Conditions

All DM and control strains were grown at 25 °C and L:D 12:12 for 7 d on the minimal medium CDA (Czapek-Dox agar: 3% sucrose, 0.3% NaNO_3_, 0.1% K_2_HPO_4_, 0.05% KCl, 0.05% MgSO_4_ and 0.001% FeSOa_4_ plus 1.5% agar) and CDAs amended with different carbon or nitrogen sources or with deleted carbon or nitrogen source. The colony growth of each strain was commenced by spotting 1 µL of a 10^6^ conidia/mL suspension per plate. As an index of radial growth, the diameter of each colony was measured perpendicularly to each other across the center. The same method was also used to initiate radial growth on the plates of CDA alone (control) or supplemented with different stress agents. The tested agents included the osmotic salt NaCl (0.7 M), the osmotic carbohydrate sorbitol (1.5 M), the oxidants menadione (0.02 mM) and H_2_O_2_ (2 mM), and the cell wall perturbing chemicals Congo red (12 µg/mL) and calcofluor white (20 µg/mL). After a 7d incubation at 25 °C, each colony diameter was measured as aforementioned. The sensitivity of each strain to each of the tested stresses was assessed as the percentage of relative growth inhibition (RGI) using the formula (*d*_c_ − *d*_s_)/*d*_c_ × 100, where *d*_c_ and *d*_s_ denote the diameters of control and stressed colonies, respectively.

### 2.4. Assessments of Conidial Yield and Quality

Since the standard medium SDAY (Sabouraud dextrose agar plus yeast extract) for fungal insect pathogens is too rich for the WT strain’s conidiation, cultures used for assessment of conidial yield per strain were initiated by spreading 100 µL aliquots of a 10^7^ conidia/mL suspension on the plates (9 cm diameter) of 1/4 SDAY amended with one-fourth (1.0% glucose, 0.25% peptone, 0.25% yeast extract and 1.5% agar) of each SDAY nutrient. During a 15-d incubation at the optimal regime of 25 °C in a light/dark (L:D) cycle of 12:12, three samples were taken from each of 8, 10 and 15 d-old plate cultures using a cork borer (5 mm diameter). Conidia in each sample were washed off into 1 mL of 0.05% Tween 80 through a 10-min supersonic vibration. The conidial concentration in the suspension was quantified using a hemocytometer and converted to an absolute yield (no. conidia/cm^2^ plate culture). The quality of conidia from the 15 d-old SDAY cultures was evaluated with two indices, namely median germination time (GT_50_) assessed at 25 °C and hydrophobicity assessed in a biphasic (aqueous-organic) system, as described previously [35].

### 2.5. Fungal Virulence Bioassays

Standardized bioassay for fungal virulence was initiated by topical application (10-s immersion in 40 mL aliquots) of a 2 × 10^7^ conidia/mL suspension to three groups of ~35 *Galleria mellonella* larvae (fifth instar) for normal cuticle infection (NCI) by each strain. Alternatively, cuticle-bypassing infection (CBI) was commenced by injecting 5 µL of a 5 × 10^4^ conidia/mL suspension into the hemocoel of each larva (~250 conidia injected) in each group. All treated groups were maintained at 25 °C and monitored daily for survival or mortality records until larval mortality reached 100% or had no more change. The resultant time-mortality trend in each group was subjected to modeling analysis for the estimation of median lethal time LT_50_ (d) as a virulence index.

Total activities (U/mL) of extracellular (proteolytic, chitinolytic and lipolytic) enzymes and Pr1 family proteases crucial for successful NCI were measured from the supernatant of each of three 3 d-old CDB (i.e., agar-free CDA) cultures containing the sole nitrogen source of 0.3% bovine serum albumin (BSA), as described elsewhere [15,20]. The submerged cultures were generated by shaking 50 mL aliquots of a 10^6^ conidia/mL suspension in CDB-BSA at 25 °C.

During the period of bioassays, hemolymph samples were taken from the larvae surviving 48–96 h post-CBI and 96–168 h post-NCI and examined to reveal a status of proliferation in vivo. Since hyphal bodies were not directly observable in the samples, 100 µL of hemolymph from three larvae (infected per strain) on each sampling occasion was added to 3 mL of SDBY (i.e., agar-free SDAY) containing kanamycin (100 ng/mL) and ampicillin (200 ng/mL) for bacterial inhibition, followed by a 36-h incubation at 25 °C on a shaking bed (150 rpm). Cells were collected by centrifugation and resuspended in 0.05% Tween 80. Samples from each suspension were then examined under a microscope to reveal the status and abundance of fungal cells and insect hemocytes.

### 2.6. Transcriptomic Analysis

Three cultures (replicates) of the Δ*cfp1* and WT strains grown for 3 d on cellophane- overlaid plates of 1/4 SDAY at the optimal regime were sent to Lianchuan BioTech Company (Hangzou, China) for construction and analysis of *cfp3*-specific transcriptomes, as described previously [35]. Clean tags generated by filtration of all raw reads from sequencing were mapped to the *M. robertsii* genome [4]. The significant levels of both log_2_ ratio (fold change) ≤−1 or ≥1 and *q* < 0.05 were used to identify down- or up-regulated genes (DEGs). All identified DEGs were enriched to GO terms in three function classes (*p* < 0.05) through Gene Ontology (GO) analysis (http://www.geneontology.org/ (accessed on 4 June 2022)) and to various pathways (*p* < 0.05) through Kyoto Encyclopedia of Genes and Genomes (KEGG) analysis (http://www.genome.jp/kegg/ (accessed on 4 June 2022)).

## 3. Results

### 3.1. Sequence Features of CFP1–3

The BLASTp search resulted in the identification of CFP1–3 homologs (EFY97868, EFY97692 and KHO10933, respectively) previously annotated as hypothetical proteins in the *M. roberstii* genome [4]. CFP1 (128 aa) is close to the query of *B. bassiana* CFP in molecular size but slightly smaller than CFP2 (141 aa) or CFP3 (145 aa). Sequence alignment analysis revealed 40–43% identities of CFP1–3 to the query (Figure 1). The three homologs share the same sequence features with the query, including no Cys residue, no predictable domain and the presence of a signal peptide (residues 1–17) at N-termini. In addition, one or two NLS motifs were predicted from the sequences of CFP1 (residues 49–78), CFP2 (residues 24–56 and 105–136) and CFP3 (residues 58–91 and 110–144). Overall, the CFP1–3 homologs in *M. roberstii* are similar to the query in molecular size and sequence feature.

### 3.2. Limited Roles of cfp1–3 in Asexual Cycle In Vitro

Compared to the WT strain, the Δ*cfp1* and Δ*cfp3* mutants showed minor defects in radial growth (suppressed by 10–20%) on CDA or CDAs amended with some of the tested carbon or nitrogen sources (Figure 2A) at the optimal regime of 25 °C and L:D 12:12. Despite little change in radial growth on the tested media, the Δ*cfp2* mutant was slightly (7%) more sensitive to osmotic stress induced by sorbitol (1.5 M) among several chemical stressors added to CDA, contrasting to a slightly increased (~9%) tolerance of Δ*cfp1* and Δ*cfp3* to cell wall stress induced by calcofluor white (20 µg/mL) and of Δ*cfp1* to oxidative stress induced by menadione (0.02 mM) (Figure 2B). All DM and control strains were equally responsive to the stresses induced by NaCl (0.7 M), Congo red (12 µg/mL) and H_2_O_2_ (2 mM). Moreover, conidial yields were measured from the cultures of 1/4 SDAY during a 15-d incubation at the optimal regime of 25 °C and L:D 12:12. As a result, conidial yields were reduced by 27–41% in Δ*cfp2* during the period but not significantly affected in the Δ*cfp1* and Δ*cfp3* mutants compared to the WT strain (Figure 2C). Median germination time (GT_50_) of conidia at 25 °C was moderately shortened in Δ*cfp2* alone (Figure 2A) while conidial hydrophobicity was lowered by ~6% in Δ*cfp2* and Δ*cfp3* (Figure 2E).

The above phenotypes were generally restored by targeted gene complementation. These data demonstrated differential, but very limited, roles for *cfp1–3* in radial growth, aerial conidiation and stress responses associated with the fungal asexual cycle in vitro.

### 3.3. Indispensable Roles of cfp1–3 in Insect Pathogenicity and Virulence

In the standardized bioassays, the control strains caused 100% mortality of *G. mellonella* larvae within 17 d after NCI or 7 d after CBI (Figure 3A). In contrast, the larvae inoculated with Δ*cfp1*, Δ*cfp2* and Δ*cfp3* for NCI still survived at 76%, 62% and 52% at the end of 17 d, respectively. All larvae inoculated with each DM for CBI also survived longer. Particularly, 37% of tested larvae survived the Δ*cfp1* CBI at the end of 10 d. Consequently, the average LT_50_ estimates were 8.35 (±0.36) d for the control strains against the model insect via NCI and 3.72 (±0.14) d via CBI (Figure 3B). However, LT_50_ was unavailable (not robustly assessable) for all DM strains (unreliable estimates: 28.3, 19.2 and 16.7 d) via NCI, and prolonged by 82% for Δ*cfp1*, 21% for Δ*cfp2* and 25% for Δ*cfp3* via CBI in comparison to the mean from the control strains.

Extracellular enzymes (ECEs) and Pr1 family proteases are collectively required for cuticle degradation during NCI [14,15]. For insight into blocked NCI of each DM, total activities of ECEs and Pr1 proteases were quantified from the supernatants of 3 d-old CDB-BSA cultures. The two quantities were significantly reduced by 58% and 15% in Δ*cfp1*, 57% and 28% in Δ*cfp2* and 34% and 5% in Δ*cfp3* relative to the WT strain, respectively (Figure 3C).

Next, we tried to directly observe the status of yeast-like budding proliferation in vivo in the hemolymph samples of infected larvae under a microscope as often observed in the samples of larvae infected by *B. bassiana* [35,36]. Unexpectedly, typical hyphal bodies were not observable in the samples of the larvae infected by control strains in either mode regardless of being stained with or without a fungal cell wall-specific dye (cottonblue or calcafluor white), suggesting a difference in the in vivo status between *M. robertsii* and *B. bassiana*. Alternatively, every 100 µL of hemolymph added to 3 mL SDBY containing kanamycin (100 ng/mL) and ampicillin (200 ng/mL) for bacterial inhibition was shaken for 36 h at the regime of 25 °C and 150 rpm. Revealed by microscopic examination of the resultant cells resuspended in aqueous 0.05% Tween 80, the WT and CM cells released from aggregated hemocytes were readily observed in the samples taken 48 h post-CBI but the Δ*cfp1* cells were not observable until 96 h post-CBI (lower panels in Figure 3D). Most host hemocytes were lysed in the samples at 48 h after CBI by the control strains but remained intact in the larvae injected with Δ*cfp1* at the same time. Moreover, the release of the Δ*cfp1* cells from host hemocytes was largely delayed in the samples of surviving larvae post-NCI and very limited even at the end of 144 h post-NCI (upper panels in Figure 3D). On the same sampling occasion, almost all released cells of the control strains were proliferating by yeast-like budding, which concurred with the disappearance of intact hemocytes. These observations implied that the fungal capability of proliferation in vivo was compromised in the absence of *cfp1*.

Altogether, three CFP genes were all indispensable for fungal insect pathogenicity and virulence via NCI, which was blocked or severely compromised in the absence of each CFP gene. The indispensability relied upon their roles in the processes of host infection through cuticular penetration and subsequent hemocoel colonization. The results also demonstrated a greater role of *cfp1* than of *cfp2* and *cfp3* in the infection cycle of *M. robertsii*.

### 3.4. Transcriptomic Insight into Indispensable Role of cfp1 in Infection Cycle

For in-depth insight into the indispensability of *cfp1* for the fungal virulence, three 3 d-old cultures (replicates) of the Δ*cfp1* and WT strains grown on cellophane-overlaid plates of 1/4 SDAY at the optimal regime were used for transcriptomic analysis. Up to 10,682 genes were detected in the transcriptome and mapped to the fungal genome [4]. Of those, 604 DEGs (up/down ratio: 251:353, the same meaning for ratios mentioned below) were identified at significant levels of log_2_ *R* ≤ −1.00 or ≥1.00 and *q* < 0.05 (Appendix A), including 154 hypothetical or functionally unknown protein genes (56:98).

The GO analysis resulted in enrichments of 711 DEGs (406:311) to 13, 83 and 50 GO terms in the function classes Cellular Component, Biological Process and Molecular Function (Appendix A), respectively. The main GO terms in the first class (Figure 4A) included cellular component (106:147), extracellular region (1:16) and extracellular space (1:4). The class also comprised more small terms, such as cell wall (1:1) and membrane raft (0:2). The low up/down ratios in these terms implied that cellular composition and membranes crucial to extracellular activity were compromised in the Δ*cfp1* mutant.

Also illustrated in Figure 4A, the main GO terms in the second class revealed the suppression of the oxidation–reduction process (15:17), proteolysis (6:8), secondary metabolic process (3:7), melanin biosynthetic process (1:4), protein catabolic process (0:5), transport (2:3) and nucleoside metabolic process (0:5). The suppressed processes were largely associated with NCI, fungus-host interaction during hemocoel colonization and production of secondary metabolites during proliferation in vivo. Interestingly, the processes related to transcription and DNA events were upregulated, including regulation of transcription by RNA polymerase II (6:4), G-quadruplex DNA unwinding (5:0), DNA duplex unwinding (5:0) and DNA replication (5:0). The main terms enriched to molecular function were catalytic activity (13:13), oxidoreductase activity (10:13), DNA binding transcription factor activity (10:3), serine-type endopeptidase activity (3:9), RNA polymerase II regulatory region sequence-specific DNA binding (6:2), phosphopantetheine binding (2:6), transferase activity (9:7), transmembrane transporter activity (2:4), telomerase inhibitor activity (5:0) and ATP-dependent 5′-3′ DNA helicase activity (5:0). In this class, notably, those terms involved in transcription and DNA events were also upregulated. The GO terms upregulated in the absence of *cfp1* suggested a requirement of certain facilitated DNA events for the mutant’s survival.

The KEGG analysis resulted in a significant enrichment of 93 DEGs (44:49) to 14 pathways (Figure 4B, Appendix A). There were 12 (6:6) DEGs enriched to glycine, serine and threonine metabolism,8 (1:7) to amino sugar and nucleotide sugar metabolism, 11 (6:5) toglycerophospholipid metabolism, 9 (7:2) to tyrosine metabolism, 8 (5:3) to phenylalanine metabolism, 7 (3:4) to fatty acid biosynthesis, 7 (4:3) to β-alanine metabolism, 5 (0:5) to galactose metabolism, 5 (3:2) to histidine metabolism, 6 (1:5) to phagosome, 4 (1:3) to cyanoamino acid metabolism, 5 (3:2) to butanoate metabolism, 3 (1:2) to prodigiosin biosyntheses and 3 (3:0) to vitamin B6 metabolism.

Illustrated in Figure 4C were the top 30 genes repressed (−16.96 ≤ log_2_*R* ≤−5.06) as if individually deleted like *cfp1* (MAA_06651, top 5). These genes encode enzymes or proteins presumably involved in cuticle degradation (MAA_06955, MAA_07484, MAA_08718 and MAA_10899), transmembrane activity (MAA_08314, MAA_09537 and MAA_10289), cell wall composition (MAA_09204 andMAA_08312), biosynthesis (MAA_00356, MAA_06744 and MAA_08699), cell cycle (MAA_08288), cellular signaling (MAA_07195), and posttranslational modifications and RNA/DNA events (MAA_00383, MAA_00423, MAA_08675 and MAA_11588), as well as nine hypothetical proteins. In contrast, most of the top 30 upregulated genes were expressed at the levels of log_2_ *R* <4 (Figure 4D) and not directly associated with the mutant’s phenotypes. So upregulated genes encode zinc finger or DNA-binding proteins (MAA_06025, MAA_10040 andMAA_10117), synthases (MAA_10033 and MAA_10202),dehydrogenases (MAA_0670 and MAA_07932), peptidases (MAA_01699 and MAA_08535), cytosine deaminase (MAA_01195), dioxygenase (MAA_08623), oxidase (MAA_10219), reductase (MAA_08621), membrane transporter (MAA_03269), protein kinase-like protein (MAA_00863). Aside from those (1/3) functionally unknown, most of the highly upregulated genes were involved in transcriptional mediation, carbon and nitrogen metabolism, stress response and signal transduction, suggesting their special roles in counteracting the negative effect of *cfp1* deletion on fungal life.

There were many more DEGs associated with or responsible for the Δ*cfp1* mutant’s main phenotypes. Listed in Appendix A were 39 DEGs (8:31) encoding varieties of hydrolases involved in cuticle degradation, 15 DEGs (3:12) involved in cell wall integrity and hydrocarbon epitopes and 43 DEGs (11:32) involved in cellular transport and homeostasis vital for manifold cellular processes and events. The majority of these genes were repressed at the transcriptional level. In addition, dozens of DEGs were involved in direct or indirect transcription mediation, including the coding genes of 22 putative transcription factors (9:13) and of 53 proteins/enzymes involved in posttranslational modifications (5:8) or RNA/DNA processing and events (20:20). However, the key developmental activator genes were not present in the list of identified DEGs, coinciding well with an insignificant change in the mutant’s conidiation level.

Altogether, our transcriptomic data revealed an important role for CFP1 in genome- wide gene expression related to a great loss of the mutant’s insect pathogenicity via NCI and a marked reduction in its virulence via CBI. Nearly two–thirds of 154 DEGs coding for hypothetical proteins were repressed, suggesting an essential role of CFP1 in transcriptional activation of those genes functionally unknown but likely crucial to host infection and hemocoel colonization of *M. robertsii*.

## 4. Discussion

Based on our experimental data, CFP1, CFP2 and CFP3 were all validated virulence factors in *M. robertsii* in comparison to the unique CFP discovered in the time-course transcriptomes of *B. bassiana* infecting *P. xylostella* [29] and characterized as sole virulence factors among 10 SSPs studied [35]. Importantly, CFP1 was closer to CFP than CFP2 and CFP3 in both molecular size and function because the *cfp1* deletion resulted in blocked NCI and greatly attenuated virulence via CBI. For this reason, the Δ*cfp1* mutant was chosen for transcriptomic analysis and comparison to the previous Δ*cfp*-specific transcriptome. While the dataset helped us understand the mutant’s main phenotypes, genome-wide gene expression was less affected by the deletion of *cfp1* in *M. robertsii* than of *cfp* in *B. bassiana* [35]. Nonetheless, the previous and present studies reinforce that CFP and its homologs in *M. robertsii* serve as extraordinary virulence factors in the wide-spectrum insect pathogens of Cordycepitaceae and Clavicipitaceae in Hypocreales, respectively.

First of all, all CFP homologs are putative SSPs due to the existence of an N-terminal signal peptide consisting of the first 17 amino acids in their sequences, but do not necessarily act as secreted proteins. This was evidenced by the fact that fluorescence-tagged CFP fusion protein was localized exclusively in the nucleus under normal conditions, partially migrated to cytoplasm in response to stress cues, but was undetectable at the extracellular level [35]. The predicted NLS motifs hint that CFP1–3 could localize in the nucleus and take part in nuclear events as did CFP in the previous study. The involvement of each CFP homolog in nuclear events was clarified by the dysregulation of 604 genes in the present Δ*cfp1* mutant and of 1818 genes (1006:801) in the previous Δ*cfp* mutant. Perhaps due to the existence of three homologs, the deletion of *cfp1* in *M. robertsii* resulted in only one–third of those dysregulated genes seen in the Δ*cfp* mutant. In particular, the counts of DEGs involved in direct or indirect genome-wide gene regulation were much smaller in the absence of *cfp1* than of *cfp*. In the Δ*cfp* mutant, for instance, there were 58 DEGs (34:24) encoding transcription factors involved in direct gene mediation and many more DEGs (126:49) encoding proteins/enzymes involved in RNA/DNA processing and events, translation, and posttranslational modifications [35]. The counts of dysregulated genes involved in nuclear events suggest a comprehensive effect of either CFP1 or CFP on the fungal insect-pathogenic lifecycle.

Aside from essential roles in insect pathogenicity and virulence, *cfp1*, *cfp2* and *cfp3* were dispensable for most of examined phenotypes in *M. robertsii*. This is largely different from the pleiotropic effects of *cfp* on the lifecycle in vitro and in vivo of *B. bassiana*, including aerial conidiation, submerged blastospore production, spore quality, and cellular responses to oxidative, osmotic, cell wall perturbing and heat stresses. Due to minor or insignificant changes in most phenotypes, counts of DEGs associated with the Δ*cfp1* phenotypes were quite limited in comparison to those involved in the previous Δ*cfp* mutant’s cell wall integrity (13:16), response to heat shock (2:6), vacuolar/cellular homeostasis and cellular transport (113:34), and signal transduction (23:9) vital for asexual development, multiple stress responses and/or virulence. The differences in both phenotypic and transcriptomic changes between Δ*cfp1* and Δ*cfp* hint that *M. robertsii* and *B. bassiana* are distinctive in genetic background and regulatory mechanisms underlying lifecycle in vitro and in vivo. Such differences have also been revealed in the previous analysis of the master transcription factor Msn2 in the two fungi [37] and in the reviews on fungal adaptation to insect-pathogenic lifecycle and host habitats [8,9,10,38,39]. The differences could be likely due to different evolutionary histories of their insect pathogenic lifestyles [3,4,5,6,7].

Finally, it remains elusive how CFP homologs enable genome-wide gene mediation. Previously, electrophoretic mobility shift assays (EMSAs) revealed no shifting signal for the binding of purified CFP samples to any samples of promoter DNA fragments of 12 genes sharply repressed in the absence of *cfp*, excluding a possibility for CFP to act as a transcription factor [35]. Thus, we did not try to repeat EMSAs in this study since both CFP1 and CFP have no predictable domain as a clue to DNA-binding activity for further exploration. Like CFP, cytoplasmic Rei1 as a pre-60S subunit export factor and nuclear Ssr4 as a co-subunit of chromatin-remodeling SWI/SNF and RSC complexes were evidently involved in the fungus-insect interaction [29] and characterized as virulence factors and genome-wide gene regulators in *B. bassiana*, but displayed no DNA-binding activity in EMSAs [33,34]. The regulatory roles of Ssr4 and Rei1 were postulated to be associated with their involvements in the SWI/SNF or RSC complex, which can bind the binding sites of transcription factors [40,41], and the dissociation and recycling of nucleocytoplasmic pre-60S factors [42,43,44], respectively. The Δ*cfp1*-specific transcriptome contained dozens of DEGs involved in direct or indirect mediation of gene expression, suggesting possible linkages of those DEGs with the regulatory role of CFP1 in *M. robertsii*. Therefore, we still speculate that small CFP homologs existing exclusively in wide-spectrum insect pathogens might be components of certain nuclear or nucleocytoplasmic protein complexes to mediate expressions of manifold genes involved in the fungal insect pathogenic lifecycle, warranting further exploration.

## Figures and Tables

**Figure 1 jof-08-00606-f001:**
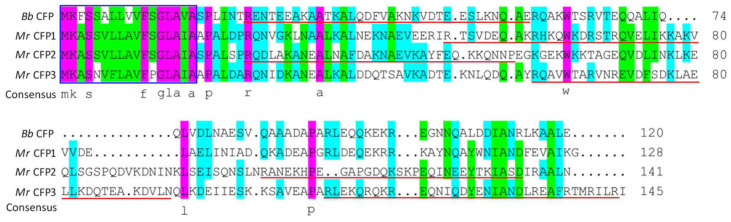
Sequence comparisons of three *M. robertsii* (*Mr*) CFP homologs (CFP1–3) with unique *B. bassiana* (*Bb*) CFP. Amino acid sequences were aligned at http://www.bio-soft.net/format/DNAMAN.htm (accessed on 4 June 2022). N-terminal signal peptide (in blue rectangle) and NLS motif (underlined) were predicted from each sequence at http://www.cbs.dtu.dk/services/SignalP-4.1 (accessed on 4 June 2022) and https://www.novopro.cn/tools/nls-signal-prediction (accessed on 4 June 2022), respectively. Note that each homolog sequence has neither Cys residue nor predictable domain.

**Figure 2 jof-08-00606-f002:**
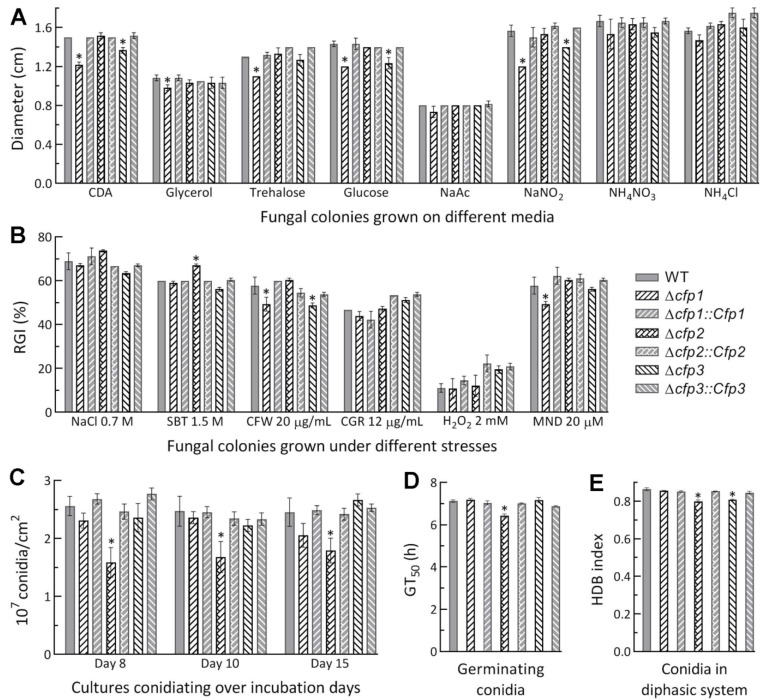
Limited roles of *cfp1–3* in the asexual cycle in vitro of *M. robertsii*. (**A**) Diameters of fungal colonies grown at the optimal regime of 25 °C and L:D 12:12 for 7 d on minimal medium CDA and CDAs amended with different carbon or nitrogen sources. (**B**) Relative growth inhibition (RGI) percentages of fungal colonies grown at 25 °C for 7 d on CDA plates supplemented with indicated concentrations of stress agents (SBT, sorbitol; CFW, calcofluor white; CGR, Congo red and MND, menadione). All colonies were initiated by spotting 1 µL aliquots of a 10^6^ conidia/mL suspension. (**C**) Conidial yields measured from the cultures during a 15-d incubation on 1/4 SDAY plates at the optimal regime after initiation of each culture by spreading 100 µL of a 10^7^ conidia/mL suspension per plate. (**D**) GT_50_ for 50% germination of conidia at 25 °C. (**E**) Indices of conidial hydrophobicity (HDB) assessed in a biphasic (aqueous-organic) system. * *p* < 0.05 in Tukey’s HSD tests. Error bars: standard deviations (SDs) from three independent replicates.

**Figure 3 jof-08-00606-f003:**
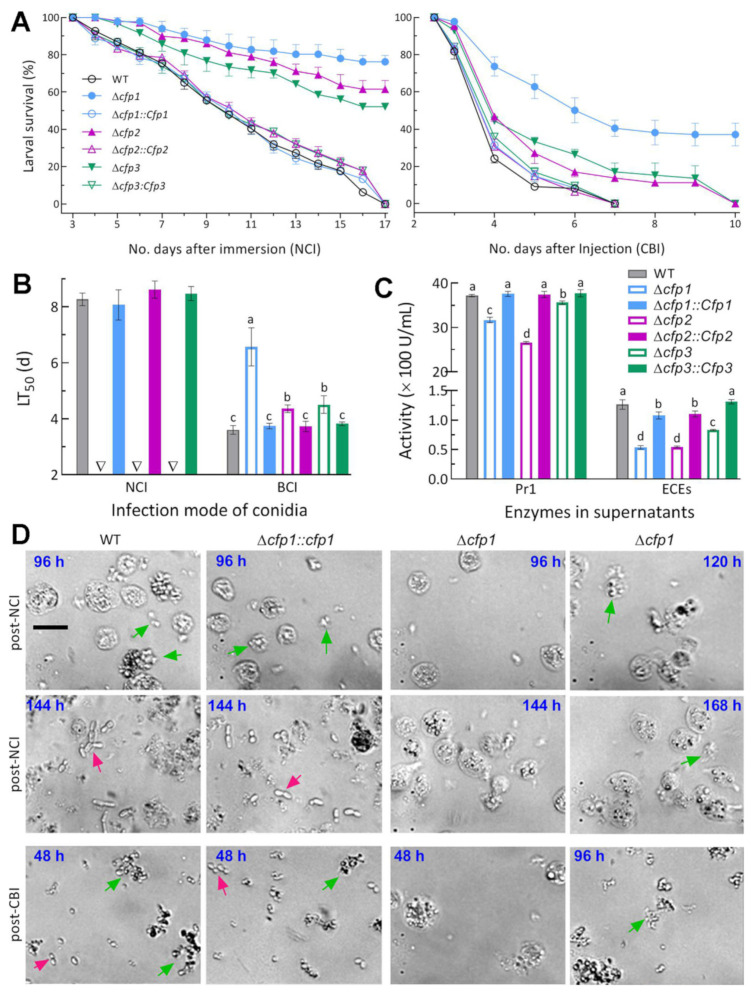
Essential roles of *cfp1–3* in the infection cycle of *M. robertsii*. (**A**) Survival trends of *G. mellonella* larvae after normal cuticle infection (NCI) via immersion in a 2 × 10^7^ conidia/mL suspension and cuticle-bypassing infection (CBI) via intrahemocoel injection of ~250 conidia per larva. (**B**) LT_50_ values estimated from time-mortality trends. Triangles indicate unavailable LT_50_s for three deletion mutants via NCI. (**C**) Total activities of extracellular enzymes (ECEs) and Pr1 proteases quantified from the supernatants of 3 d-old submerged cultures, which were generated by shaking incubation of a 10^6^ conidia/mL suspension in CDB-BSA. (**D**) Microscopic images (scale bar: 20 µm) for status and abundance of fungal cells (arrowed) and insect hemocytes (spherical or subspherical cells) in hemolymph samples, which were taken from surviving larvae at different time points after NCI (upper) and CBI (lower) and incubated at 25 °C for 36 h in SDBY. Green arrows indicate fungal cells released from aggregated hemocytes. Red arrows indicate yeast-like budding proliferation of fungal cells released. Different lowercase letters in (**B**,**C**) indicate significant differences (Tukey’s HSD, *p* < 0.05). Error bars: SDs from three independent replicates.

**Figure 4 jof-08-00606-f004:**
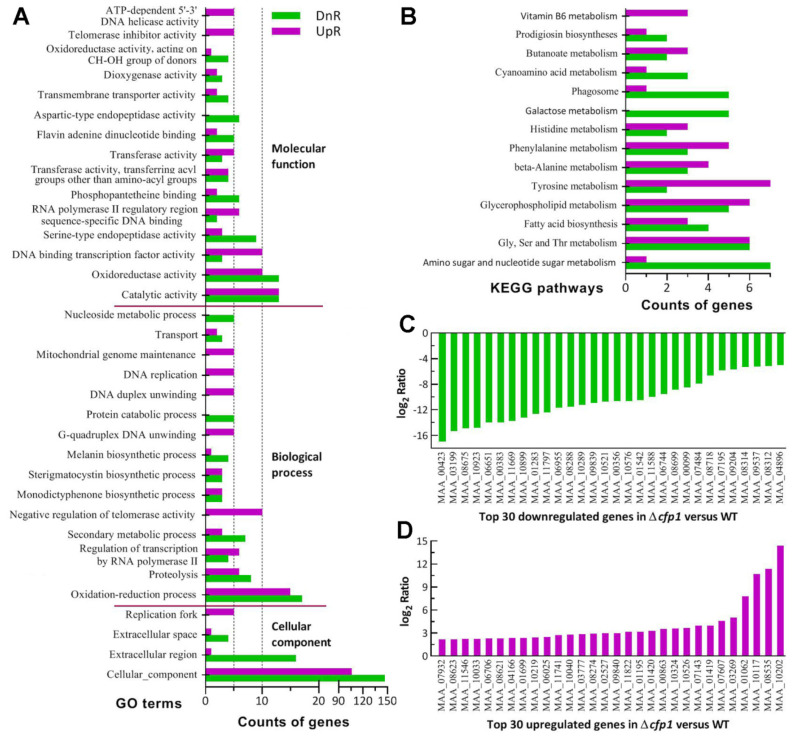
Effect of *cfp1* deletion on genome-wide gene expression of *M. robertsii*. (**A**) Counts of differentially expressed genes (DEGs) significantly enriched (*p* < 0.05) to three function classes (main GO terms shown; see Appendix A for all GO terms). (**B**) Counts of DEGs significantly enriched (*p* < 0.05) to KEGG pathways. The transcriptome was constructed based on three 3 d-old cultures (replicates) of the Δ*cfp1* and WT strains. All DEGs were identified at the significant levels of log_2_ ratio (Δ*cfp1*/WT) ≤ −1 (downregulated, DnR) or ≥1 (upregulated, UpR) and *q* < 0.05. (**C**,**D**) The log_2_ ratio values of top 30 down- and upregulated genes.

## Data Availability

All experimental data are included in this paper and Appendix A. All RNA-seq data are available at the NCBI’s Gene Expression Omnibus under the accession GSE197269 (https://www.ncbi.nlm.nih.gov/geo/query/acc.cgi?ac=GSE197269 (accessed on 4 June 2022)) aside from those reported in Appendix A in Appendix A of this paper.

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
