# Peer review of "Three Small Cysteine-Free Proteins (CFP1–3) Are Required for Insect-Pathogenic Lifestyle of Metarhizium robertsii"

_jof, 2022, doi:10.3390/jof8060606_

Round 1

Reviewer 1 Report

The manuscript “Three Small Cysteine-Free Proteins (CFP1–3) Are Required for 2 Insect-Pathogenic Lifestyle of Metarhizium robertsii” by Mou and coworkers describes the functional characterization of 3 cysteine-free proteins and the impact of these deletions on Metarhizium’s lifecycle and virulence. The manuscript is good and the experiments well-thought-out, but it also needs some improvements.

Minor reviews:

English proofreading by a native speaker should be considered.

As far as I know, there is no difference between “virulence factor” and “virulence effector”. Thus, the authors must adopt only one term.

Line 72: nonentomopatho- genic

Line 149: Fungsl

Major reviews:

It is not clear how many transformants were isolated for each gene. Moreover, do all isolated transformants display the same phenotype? Given the overall importance of the cfp genes, as described by the authors, secondary mutations caused by the deletion cassette is a matter that worries me. Did the authors explore if the cfp genes were the only ones mutated in the strains? Additionally, I suggest Southern blot to double-check the generated transformants.

All assays conducted, but the RNA-seq analysis, must be performed with at least 3 different transformants for each gene. It not clear if the assays were conducted in this way.

Lines 159-163: A better and more detailed description of the extracellular enzyme assay must be provided. There is no clue about how the authors performed such experiment.

There is no description of Statistics in the Material and Methods section, a subsection must be provided.

I do not like how the hemocytes experiment was performed. I do not see a reason to perform the experiment in this way (incubating the fungal cells and hemocytes in SDBY), even with the authors' explanation. I never performed this experiment, but if other authors performed it, it should be possible to be done in the right way. I would recommend the manuscript “Unveiling the biosynthetic puzzle of destruxins in Metarhizium species” since a similar experiment was described. Moreover, the image quality of Figure 3D must be improved, there is no control (hemocytes only) as well. The results for cfp2 and cfp3 should be also amended.

Line 282: The authors only performed virulence assays employing Galleria mellonella, although the results pointed to a potential impact of those genes on the virulence of M. robertsii, it is a bit harsh to say that these genes affect “the infection cycle of M. robertsii” as a general statement. It would be interesting to explore how these deleted strains perform in virulence assays against other hosts. Preferably, against non-Lepidoptera hosts.

Lines 331-336: A better description of these genes would be nice. Did the authors find characterized genes down-regulated?

How do the authors explain that an apparent master-regulator of entomopathogenic fungi is not conserved in M. acridum or M. rileyi? did they lose these genes through evolution? Annotation problems? Are these genes conserved in other non-insect pathogens of Hypocreales order?

Figure S1 must be improved. Not sure what the letters in the legend state.

Author Response

Please see attached  a file.

Reviewer 2 Report

This is an interesting study. The paper is generally well written and structured. The manuscript provides a valuable idea but requires minor corrections before publication:
•    Check all scientific names should be written in italic form
•    Line 7: CFP (Cycteine-Free Protein; 120 aa) should be Cycteine-Free Protein; 120 aa (CFP)
•    Line 48: add reference at the end of the paragraph
•    Line 77-80: rephrase in more clear method
•    Keywords: arrange keywords alphabetical and capitalized the start of each word.
•    Material and methods include too much information, try to reduce its contents.
•    Results: figure 1: try to increase its quality
•    Line 412: add this part under subtitle conclusion
•    Reference: check references within the manuscript and in the reference section according to journal guidelines.

Author Response

Please see attached a file.
